**TECHNIQUE**

# Feasibility of multimodal magnetic resonance imaging to assess maternal hyperoxygenation in sheep pregnancy

Dimitra Flouri[1,2] , Jack R. T. Darby[3] , Stacey L. Holman[3] , Georgia Williams[4] ,
Vasileios Vavourakis[1,5] , Anna L. David[6] , Janna L. Morrison[4] and Andrew Melbourne[2,6]

[1] *In Silico Modelling Group, Department of Mechanical & Manufacturing Engineering, University of Cyprus, Nicosia, Cyprus*
[2] *School of Biomedical Engineering & Imaging Sciences, King's College London, London, UK*
[3] *Early Origins of Adult Health Research Group, Health and Biomedical Innovation, UniSA: Clinical and Health Sciences, Adelaide, SA, Australia*
[4] *Preclinical Imaging and Research Laboratories, South Australian Health and Medical Research Institute, Adelaide, SA, Australia*
[5] *Department of Medical Physics & Biomedical Engineering, University College London, London, UK*
[6] *Elizabeth Garrett Anderson Institute for Women's Health, University College London, London, UK*

Handling Editors: Kim Barrett & Laura Bennet

The peer review history is available in the Supporting Information section of this article
(https://doi.org/10.1113/JP287272#support-information-section)

**Abstract figure legend** Magnetic resonance imaging (MRI) techniques are increasingly providing information on placental function *in vivo* to support clinical decision making. Preclinical models such as pregnant sheep have been important in invasive validation studies for MRI measurements because they allow for controlled experiments and analysis at multiple time points during pregnancy. Here, the physiological impact of maternal hyperoxygenation on the placenta in normal pregnant sheep was investigated using multimodal functional MRI. Maternal hyperoxygenation resulted in a significant increase in blood-oxygenation-level-dependent (BOLD) signal intensity, as well as an increase in feto-placental oxygen saturation.

J. L. Morrison and A. Melbourne contributed equally to this work.

**Abstract**  An adequate supply of oxygen is crucial for optimal fetal growth and development. Estimation of quantitative indices that reflect tissue diffusivity and oxygenation have been enabled by advances in magnetic resonance imaging (MRI) technology. However, the current diagnostic tools in clinical obstetrics, such as Doppler ultrasound measurements of umbilical blood flow and cardiotocography, do not offer direct information about the oxygen supply to the fetus, nor placental function *in vivo*. Although MRI provides an opportunity to identify critical changes in fetal oxygenation, exact tissue oxygen content cannot be established in humans. Preclinical models such as pregnant sheep allow the use of invasive methods to validate MRI measurements. The present study investigates the relationship between changes in MRI signal and conventional blood gas analyser measurements during normoxic and hyperoxic conditions in pregnant sheep. Several studies have reported an increase in human fetal oxygenation during 100% maternal oxygen inhalation. We investigated the physiological impact of maternal hyperoxygenation on the placenta in normal pregnant sheep using multimodal functional MRI. Using a multicompartment MRI signal model, we observed the expected increase in feto-placental oxygen saturation with maternal hyperoxygenation. In addition, maternal hyperoxygenation resulted in a significant increase in blood-oxygenation-level-dependent (BOLD) signal intensities, suggesting that BOLD MRI allows non-invasive assessment of the feto-placental response to maternal hyperoxygenation in sheep. Our data suggest that diffusion and relaxation-based MRI is sensitive to acute changes in maternal and feto-placental oxygenation and demonstrate a link between MRI-parameter estimated and reference oxygen saturation.

(Received 30 July 2024; accepted after revision 21 January 2025; first published online 12 February 2025)

**Corresponding author** D. Flouri: In Silico Modelling Group, Department of Mechanical & Manufacturing Engineering, University of Cyprus, Nicosia GP505, Cyprus. Email: flouri.dimitra@ucy.ac.cy

**Key points**

- Quantification of feto-placental oxygenation and function are important for correct differential diagnosis of placental insufficiency. The only current method for obtaining information about fetal oxygen delivery is cordocentesis. However, there is a risk of inducing preterm birth and/or fetal loss associated with the procedure.
- Magnetic resonance imaging (MRI) can estimate changes in oxygenation in specific areas of placental and fetal tissue.
- Using the DECIDE (i.e. diffusion-relaxation combined imaging for detailed placental evaluation) multicompartment model that is sensitive to changes in maternal and feto-placental oxygenation and the blood-oxygenation-level-dependent (BOLD) MRI technique in the sheep fetus, we have demonstrated that maternal hyperoxygenation increases oxygenation of fetal tissue in the placenta. There was a differential effect according to placentome morphological type.
- This study shows a link between MRI estimated parameters and reference maternal and fetal $S_{O_2}$ and $P_{O_2}$ by blood gas analyser, supporting the possibility of using multimodal MRI for measuring regional changes in tissue oxygenation *in vivo*.

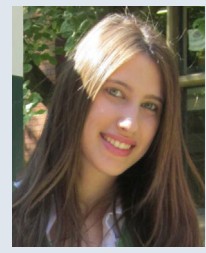

**Dimitra Flouri** earned her PhD in Biomedical Imaging and Applied Mathematics at the University of Leeds in UK. She is a recipient of a European Marie-Curie Postdoctoral Fellowship. Her research interests fall primarily in the areas of image processing and quantitative magnetic resonance imaging. Her current research focuses on the development and application of advanced medical imaging techniques that provide a more accurate and biologically specific assessment of placental perfusion, function and microstructure.

## Introduction

An adequate supply of oxygen is crucial for fetal growth and development, but the fetus relies on the placenta for this supply. Oxygen is delivered to the placenta via maternal blood flow predominantly through the uterine arteries and crosses the placenta between the separate maternal and fetal circulations (Carter, 1999; Goplerud & Delivoria-Papadopoulos, 1985; Saini et al. 2021). Reduced fetal oxygen transfer from the maternal bloodstream occurs in cases of placental dysfunction, which is associated with hypoxaemia (Soares et al. 2017), fetal growth restriction (Sun et al. 2020), pre-eclampsia and adverse perinatal outcome (Baschat, 2011).

Ultrasound measurement of fetal size, Doppler ultrasound measurements of umbilical blood flow and cardiotocography are the current diagnostic tools to estimate fetal well-being. Antenatal computerised cardiotocography permits the calculation of beat-to-beat variation, termed short-term-variation, which, if low (<3.0 ms), is correlated with stillbirth and severe birth academia (Dawes et al. 1992; Galazios et al. 2010; Wretler et al. 2016). However, these methods do not offer direct information about the oxygen supply to the fetus, nor placental function *in vivo* (Clark et al. 2018). The only current method for obtaining direct information about fetal oxygen delivery is ultrasound-guided umbilical vein blood sampling via cordocentesis. However, this invasive prenatal procedure is not routinely used to assess feto-placental dysfunction in clinical obstetrics because of the perceived risk of inducing preterm birth or fetal loss (Salomon et al., 2019; Zhen & Li, 2020).

With magnetic resonance imaging (MRI), it is now possible to estimate changes in oxygenation in specific areas of placental and fetal tissue. The blood-oxygenation-level-dependent (BOLD) effect in MRI is based on the magnetic properties of haemoglobin. Deoxygenated haemoglobin has paramagnetic properties (Prasad et al., 1996); therefore, changes in the blood saturation will affect the transverse relaxation time (T2 and T2$^*$) of the surrounding water molecules, providing a measurable change in the MRI signal on both T2 and T2$^*$ weighted imaging. T2$^*$-weighted sequences have proved to be a simple but useful method to assess placental function, either by directly measuring the value of T2$^*$, or by assessing the change of relative signal intensity of the raw T2$^*$-weighted MR images (Bartin et al., 2024; Hutter et al., 2019; Sørensen et al., 2020).

Although static T2$^*$ mapping may be used as an indirect marker for quantifying oxygenation, repeatedly acquiring T2$^*$ weighted images with a repetition time of a few seconds allows the assessment of dynamic signal change with intervention such as maternal hyperoxia. This repeated imaging can directly assess changes in oxygen transport within the placenta and has emerged as a promising tool to study placental function that has been validated in both humans and sheep (Abaci et al., 2019; Sørensen et al., 2013). In humans, dynamic BOLD MRI during maternal hyperoxygenation identified the effect of maternal position and uterine contractions on maximum signal change in the placenta (Abaci et al., 2020). Thus, BOLD MRI represents an interesting new contrast for placental studies, where the uptake rate of oxygen may relate to gestational or pathological tissue oxygenation changes.

BOLD signal change in the placenta is affected by maternal blood in the intervillous space and fetal blood in the fetal capillaries (Egbor et al., 2006). Because neither compartment is fully saturated by oxygen under ambient conditions (80% and 60%, respectively) (Sun et al., 2015) when the mother inspires 100% oxygen, deoxygenated haemoglobin content drops, causing an increase in BOLD signal as a result of a longer blood T2$^*$. The feasibility of the BOLD effect to reflect change in oxygen saturation has been demonstrated in rat (Aimot-Macron et al., 2013; Chalouhi et al., 2013), macaque (Schabel et al., 2016) and human (Avni et al., 2016; Sinding et al., 2016; Sørensen et al., 2013, 2015) placenta. However, a limitation of this approach is that the BOLD signal intensity values are measured in arbitrary units, which allows changes in relative oxygenation to be observed, but not the measurement of absolute oxygen saturation. In addition, T2$^*$ weighted image contrast also depends upon the local susceptibility differences of tissue compartments. As a result, the effect on contrast depends on the surrounding tissue and is thus non-compartmental and not directly related to oxygen saturation of either the fetal or maternal blood compartments.

Diffusion-weighted imaging (DWI) (Aughwane et al., 2020; Flouri et al., 2020; Kristi et al., 2020; Li et al., 2023; Lu et al., 2022) provides a non-invasive method of measuring tissue properties related to flow and perfusion when combined with intravoxel incoherent motion modelling (IVIM) (Flouri et al., 2022; Melbourne et al., 2019). T2 relaxometry is based on acquiring images at different echo times and has been used to estimate oxygen saturation in placenta (Flouri et al., 2022; Liu et al., 2022; Malmberg et al., 2022; Melbourne et al., 2019) with the stronger assumption that the signal is compartmental compared to T2$^*$. Recent advances in placental DW-MRI allow more quantitative analysis of functional properties and oxygenation and can be measured by forming a multicompartment model of the placenta (Melbourne et al., 2019). By contrast to more common applications of IVIM, placental IVIM is complicated by the close proximity of two circulations, from maternal and fetal blood within an imaging voxel, implying an additional compartment over the standard IVIM model.

In the present study, we investigated the effects of maternal hyperoxia on measurements from advanced MRI including BOLD MRI, T2* relaxation and combined diffusion with T2 relaxometry imaging (diffusion-relaxation combined imaging for detailed placental evaluation; DECIDE) (Melbourne et al., 2019). DECIDE allows modelling of human placental perfusion from separate fetal and maternal signal contributions and has previously been validated in sheep where different placentome types can be visualised (Flouri et al., 2022; Melbourne et al., 2019). The sensitivity of the DECIDE model to changes caused by maternal hyperoxia has not yet been shown; thus, here, we combine this information with data from BOLD MRI and T2* relaxometry to investigate changes in sheep placental oxygenation during maternal hyperoxia. In sheep, exchange of oxygen and nutrients between the ewe and her fetus occurs in many placentomes, which are classified into four morphological types that change with gestational age (A–D) (Ward et al., 2006). These placentomes can be categorised from MRI images (Flouri et al., 2021) and hence differences in structure and function can be studied. Therefore, the aim of the present study was to investigate the effect of maternal hyperoxygenation on oxygen saturation in the placenta using multimodal functional MRI.

## Methods

### Ethical approval

The experimental protocols were reviewed and approved by the Animal Ethics Committee of the South Australian Health and Medical Research Institute (SAHMRI) and abide by the Australian Code of Practice for the Care and Use of Animals for Scientific Purposes (2013) developed by the National Health and Medical Research Council and followed the ARRIVE guidelines (Percie et al., 2020). All investigators understood the ethical principles outlined in Grundy (2015) and the principles of the 3Rs (Steinmeyer et al., 1995).

### Animals and surgery

Ewes were sourced from the SAHMRI farm (Burra, SA, Australia) and housed in an indoor facility under a 12:12 h light/dark photocycle at a constant ambient temperature of 20–22°C and. Ewes were housed in individual pens in view of other sheep and had *ad libitum* access to food and water. Singleton-bearing Merino ewes ($n = 8$) underwent fetal catheterisation surgery at 116–117 days of gestation (term = 150 days) as previously described (Darby et al., 2018; Morrison et al., 2007). All ewes received the analgesic meloxicam ($0.5$ mg kg$^{-1}$, s.c.) on the day before surgery and 24 h later (Varcoe et al., 2019). General

anaesthesia was induced with intravenous diazepam ($0.3$ mg kg$^{-1}$) and ketamine ($7$ mg kg$^{-1}$) and maintained with isoflurane (1.5–2.5% in 100% oxygen). Vascular catheters were implanted into the maternal jugular vein, fetal femoral vein, femoral artery (tip in the descending aorta at the level of the renal artery) and the amniotic cavity as previously described (Morrison et al., 2007). The fetus was returned to the uterus, which was sutured closed. A small incision was made in the ewe's flank, allowing exteriorisation of the fetal catheters. After closure of the abdomen, ewes received an intramuscular injection of antibiotics [3.5 mL of Duplocillin (150 mg mL$^{-1}$ procaine penicillin and 112.5 mg mL$^{-1}$ benzathine penicillin; Norbook Laboratories Ltd, Gisborne, Australia) and 2 mL of 125 mg mL$^{-1}$ dihydrostreptomycin (Sigma, St Louis, MO, USA)] for 3 days following surgery. Each fetus received intra-amniotic antibiotics (500 mg; sodium ampicillin, Commonwealth Serum Laboratories) for 4 days post-surgery. At 124–125 days of gestation, ewes were humanely killed with an overdose of sodium pento-barbitone (Randlab Australia Pty Ltd, Revesby, NSW, Australia). The fetus was delivered via hysterotomy and weighed before tissues were collected for use in subsequent studies.

### Blood sampling

After fetal surgery, fetal arterial blood samples were collected daily and during the maternal normoxia and hyperoxia states of the MRI sessions to monitor fetal health by measuring the partial pressure of oxygen ($P_{O_2}$), partial pressure of carbon dioxide ($P_{CO_2}$), oxygen saturation ($S_{O_2}$), pH, haemoglobin, haematocrit, base excess and lactate with a RAPIDPOINT 500 blood gas analyser (BGA) (Siemens Healthineers, Erlangen, Germany), temperature corrected to 39°C.

### Imaging procedure

**Animal preparation.** At 123–125 days of gestation pregnant ewes ($n = 8$) underwent MRI scans. General anaesthesia was induced in the ewe as described for surgery above. The ewe was then positioned on its left side for the duration of the scan. Ventilation parameters were titrated to best match the normal fetal oxygenation prior to induction of anaesthesia (respiratory rate 16–18; ~1–2 L of O$_2$ and 4–5 L of air). Maternal heart rate and arterial $S_{O_2}$ were measured using a MRI-compatible pulse oximeter (Nonin Medical Inc., Plymouth, MN, USA). The sensor was placed on the pregnant ewe's teat and measurements were continuously recorded using LabChart 7 (AD Instruments, Bella Vista, NSW, Australia) (Darby et al., 2019; Duan et al., 2019; Schrauben et al., 2019). MR imaging studies were performed on a 3T Skyra

Scanner (Siemens Healthineers). Three 18-channel body flex coils were positioned over the ewe and were employed in conjunction with the spine matrix coil. Each imaging protocol consisted of a normoxia period and a hyperoxia period. Maternal hyperoxygenation was induced by increasing the oxygen ratio in the ventilated gas mixture (100% $O_2$, 6 L) and confirmed with maternal and fetal BGA results. All fetal blood gases taken during the MRI were arterial except for one fetus whose arterial catheter was blocked. For this animal, fetal blood gas samples were taken from the venous catheter in both oxygenation states; thus, these data are excluded from grouped means.

**Diffusion-weighted imaging and T2-relaxometry.** DWI was performed at ten $b$ values $b = [0, 10, 20, 30, 50, 70, 100, 200, 300, 500$ and $600]$ s mm$^{-2}$ and echo-time ($T_E$) = 72 ms. Spin-echo T2-relaxometry was acquired at $b$ value = 0 s mm$^{-2}$ and at 10 echo times, $T_E = [81, 90, 96, 120, 150, 180, 210, 240, 270$ and $300]$ ms. In addition, data acquired at $b$ values 50 and 200 for $T_E = [81, 90, 120, 150, 180, 210$ and $240]$ ms. Data were acquired with a pulse gradient spin-echo with echo planar imaging (EPI) readout sensitive to diffusion in the slice plane with the following parameters: flip angle = 90°, voxel size $0.9 \times 0.9 \times 6$ mm$^3$, with field of view $350 \times 281$ mm$^2$ and 26 slices.

**Bold MRI.** BOLD imaging was performed with EPI free induction decay (FID) sequence with the following parameters: repetition time = 3500 ms, $T_E$ = 35 ms, flip angle = 90°, slice gap of 0.3 mm, field-of view $400 \times 245$ mm$^2$ with voxels $2 \times 2 \times 3$ mm$^3$. In total, 172 dynamics were acquired in each 10 min BOLD scan.

**T2*-weighted imaging.** T2* images were generated from a multi-echo EPI FID pulse sequence with the following parameters: repetition time set at 3500 ms, $T_E = [26, 48, 58$ and $78]$ ms, flip angle = 90°, field-of-view $400 \times 245$ mm$^2$ with $2 \times 2 \times 3$ mm$^3$ voxels.

## Imaging data analysis

**DECIDE multicompartment model.** The DECIDE model is a multicompartment sheep-specific signal model of placental perfusion that combines DWI and T2-relaxometry (Melbourne et al., 2019). The DECIDE signal model is of the form:

$$S(b, T_E) = S_0 \left[ e^{-bd^*} \left( f e^{-T_E R_2^{f_b}} + v e^{-T_E R_2^{mb}} \right) + (1 - f - v) e^{-bd - T_E R_2^{ts}} \right] \quad (1)$$

where $S$ is the measured MR signal and $S_0$ is the signal with no diffusion weighting (i.e. $b = 0$). The five independent model parameters are the feto-placental blood volume fraction $f$, the trophoblast diffusivity $d$, the pseudo-diffusivity $d^*$, the feto-placental blood relaxation $R_2^{f_b} = 1/T_2^{f_b}$ and the maternal blood volume fraction $v$. We used literature-based values for highly saturated maternal blood relaxation $R_2^{mb}$ and tissue relaxation $R_2^{ts}$ at 3T of $(150$ ms$)^{-1}$ and $(42$ ms$)^{-1}$ (de Bazelaire et al., 2004; Stanisz et al., 2005).

**Estimation of feto-placental blood oxygen saturation.** MRI can determine $S_{O_2}$ using *in vivo* measurement of the signal decay $T_2$ and an *in vitro* calibration curve relating $T_2$ and $S_{O_2}$ based on a previously proposed equation (Saini et al., 2020; Wright et al., 1991) and as described in Aughwane et al. (2020):

$$\frac{1}{T_2} = \frac{1}{T2,0} + K_0 \left( 1 - \frac{SO_2}{100\%} \right) + K_1 \left( 1 - \frac{SO_2}{100\%} \right)^2 \quad (2)$$

where $T_2$ is the measured $T_2$ value of partially oxygenated blood, $T_{2,0}$ is the $T_2$ value of fully oxygenated blood, and $K_0$ and $K_1$ are calibration factors. $T_{2,0}$, $K_0$ and $K_1$ are held fixed at literature values of 148.4 ms, 1.4 s$^{-1}$ and 104.4 s$^{-1}$, respectively (Portnoy et al., 2018).

**DECIDE model fitting.** The Levenberg–Marquart method was applied to eqn (1) to perform voxel-based model fitting using an inhouse software developed in MATLAB (The MathWorks, Natick, MA, USA). The fitting routine initialised with parameter estimates from model-fitting results obtained from average placental region of interest (ROI) signal curves (Melbourne et al., 2019). To stabilise the fitting when computing voxelwise estimates, the following constraints were chosen: $0 < f < 1$ (no units), $0 < d < 1$ (mm$^2$ s$^{-1}$), $0 < d^* < 1$ (mm$^2$ s$^{-1}$), $0 < T_2^{f_b} < 150$ and $0 < v < 1$ (no units).

**T2 relaxometry.** The simplest model for analysing MR data considers a one-compartment model. In T2-relaxometry, the parameter of interest is the T2 relaxation rate itself and is given by the following mono-exponential decay:

$$S(T_E) = S_0 e^{-T_E/T2} \quad (3)$$

where $T_E$ is the echo time and T2 is the relaxation time.

**T2* mapping.** In addition to calculating the mean T2* relaxation time, a map of the T2* value of each pixel can be created to illustrate the T2* heterogeneity. In the present study, T2* maps were computed by log-linear voxel-wise fitting to the exponential decay curve at the four $T_E$ using the signal model:

$$S(T_E) = S_0 e^{-T_E/T2^*}, \quad (4)$$

where $S_0$ is the signal with zero diffusion weight and $T2^*$ is the relaxation time. The procedure was performed under maternal normoxia and repeated after maternal hyperoxia to assess $T2^*$ change with hyperoxygenation.

## BOLD effect

The BOLD signal of each placentome ROI was recorded during the entire 10 min BOLD scan and for each ROI the BOLD signal at each time point was normalised using the mean BOLD signal of the initial 2 min of normoxia as a reference (Fig. 1). To investigate the placentome response to maternal hyperoxia a sigmoid function was fitted to the BOLD data using a non-linear least square regression, modelling by the following equation with three free parameters for the baseline signal, $S_0$, the magnitude of signal change, $a$, rate of uptake, $b$, and time to reach half maximum, $c$:

$$Snorm\ (t) = \frac{a}{1 + e^{-b(t-c)}} + S_0 \qquad (5)$$

**Normalisation of BOLD signal.** The BOLD signal of ROI was obtained in multiple slices for each ewe at 172 time points during the entire 10 min BOLD scan. For each ROI, the BOLD sigal at each time point was normalised using the mean BOLD signal of the initial 2 min of normoxia ($BOLD_{Normoxia}$) as reference. Furthermore, $\Delta$ BOLD was calculated using:

$$\Delta BOLD\ (\%) = \left( \frac{BOLD_{Hyperoxia} - BOLD_{Normoxia}}{BOLD_{Normoxia}} \right)$$
$$\times\ 100\% \qquad (6)$$

where $BOLD_{Hyperoxia}$ is the signal intensity during maternal hyperoxia.

## Image processing

The placentome ROIs were manually segmented from the first $b = 0$ image (ITK-SNAP, version 3.6.0; http://www.itksnap.org). To avoid non-placentome tissue, the ROIs were placed away from the edges such that any residual motion artefact as a result of maternal breathing

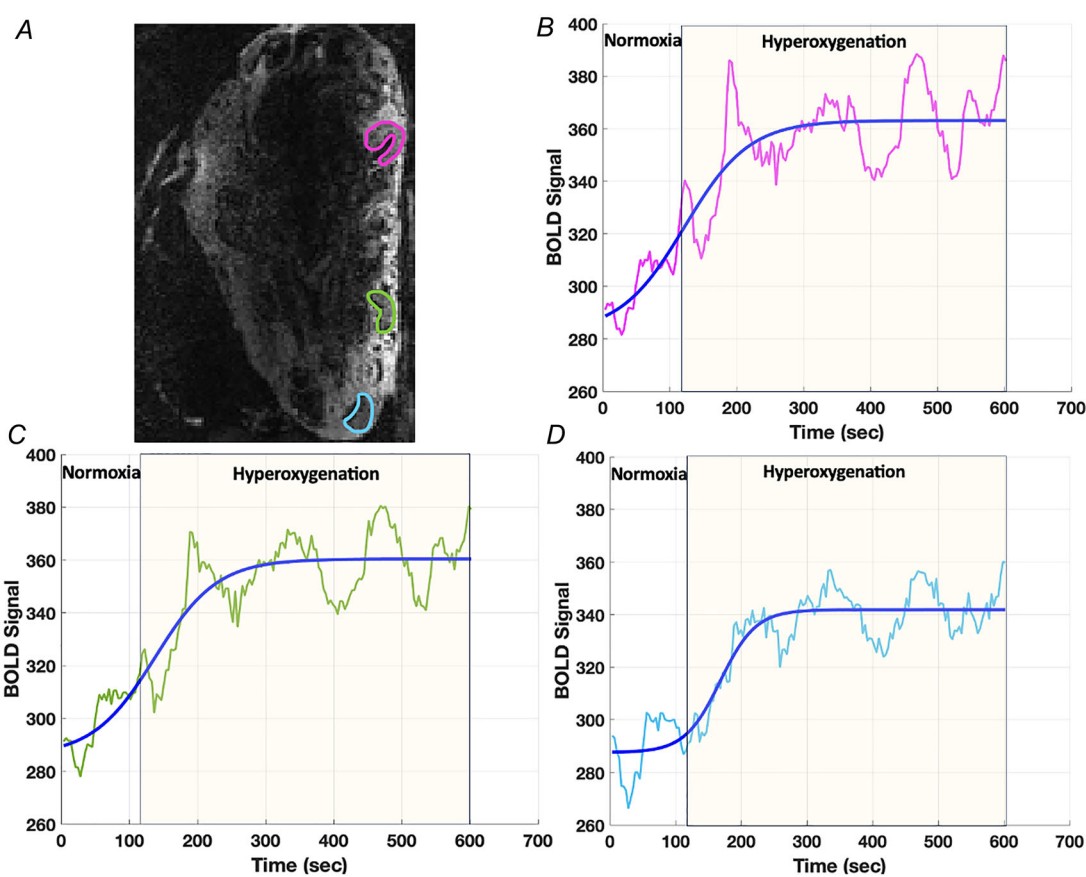

**Figure 1. Blood oxygen level-dependent (BOLD) image and Signal curves**
*A*, Placentome regions are shown in the inset BOLD image. *B-D*, Showing the blood-oxygenation-level-dependent (BOLD) signal (magenta, green, sky blue lines) *vs*. time curves of three different placentomes during 10 min BOLD scans and the corresponding sigmoid fitting (blue lines).

and fetal movement would not move the ROI out the placentome. The ROIs covered the largest area common to all time points of the dynamic acquisition. Placental MRI was used to morphologically classify placentomes into two types (A and B) using the classification system defined in a previous study (Ward et al., 2006) in which we investigated the reproducibility of placentome classification between different types. Type A and B placentomes dominate in normal pregnancy, in contrast to type C and D placentomes that are less common. Placentomes concave in shape with the maternal tissue (lighter grey-coloured) surrounding the fetal tissue (black) were classified as type A placentomes (Fig. 1*A*). Type B and C placentomes are intermediate in shape. Type B placentomes consist of fetal tissue beginning to grow over the surrounding maternal tissue and type C placentomes consists of a larger portion of fetal tissue that has begun to surround maternal tissue. Type D placentomes contain mostly fetal tissue which surrounds the maternal tissue (Flouri et al., 2021). We applied a rigid registration (Klein et al., 2010) followed by non-rigid free-form registration (Flouri et al., 2020) to reduce motion in DWI and T2-relaxometry data. To mitigate motion in the BOLD data, we used a data-driven registration method based on principal component analysis (Melbourne et al., 2007).

### Statistical analysis

Statistical analysis was performed using RStudio, version 4.0.3 (Posit PBC, Boston, MA, USA). Normality was assessed with Shapiro–Wilk test. A paired sample *t* test was performed to examine the effect of maternal hyperoxygenation on the MRI-derived parameters. Simple linear regression was performed using Prism, version 9.2.0, 2021 (GraphPad Software Inc., San Diego, CA, USA) to compare the relationship between the observed MRI measurements against the reference maternal and fetal measurements measured by conventional BGA. The correlations between recorded MRI measurements and the reference were assessed using Pearson's correlation coefficient (*r*). A two-way ANOVA was performed in RStudio 2022.07.2 (Posit PBC) to examine the effect of maternal hyperoxygenation and placentome type on MRI parameters. Numerical results are expressed as the mean ± SD. $P < 0.05$ was considered statistically significant.

## Results

### Changes with maternal hyperoxia on T2 and diffusion weighted DECIDE MRI

Figure 2 shows quantitative parameter maps obtained from T2 relaxometry, T2* mapping and the DECIDE sheep multicompartment model under conditions of maternal normoxia and hyperoxia. During hyperoxia, a clear visual change occurred in the MRI parametric maps. The placentome structures appeared brighter and homogenous.

Table 1 presents the average DECIDE parameters over the eight singleton pregnancies. Maternal hyperoxia had a significant impact on the placentome T2* and T2 values. The mean placentome T2* and T2 increased after maternal oxygen administration, and this is confirmed by the quantitative analysis of the T2 and T2* mapping that were significantly higher during maternal hyperoxia (Δ T2 = 13.61 ms, Δ T2* = 7.85 ms) (Table 1). During maternal hyperoxia, there was a 14% increase in

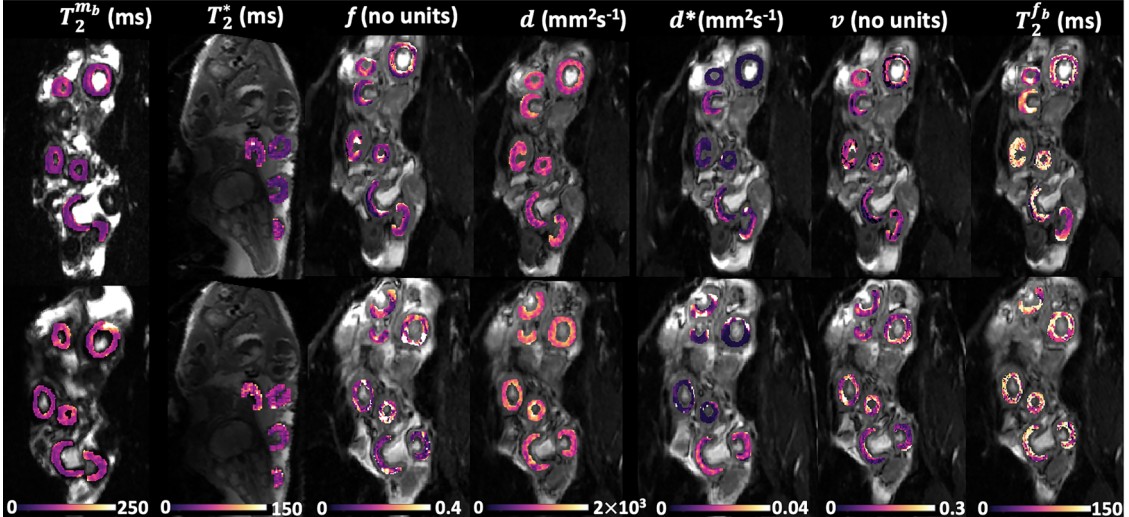

**Figure 2. Example MRI parameter maps derived from the T2\* mapping and the DECIDE sheep signal-model from one ewe**
MRI parameter maps are shown for maternal normoxia (top row) and maternal hyperoxia (bottom row).

**Table 1. Average MRI parameters, fetal and maternal venous BGA measurements derived over all singleton pregnancies during maternal normoxia and maternal hyperoxia**

| MRI parameter | Maternal normoxia ($n = 8$) | Maternal hyperoxia ($n = 8$) | P value |
|---|---|---|---|
| T2 (ms) | $127.933 \pm 7.054$ | $141.544 \pm 6.760$ | **<0.001** |
| T2* (ms) | $51.070 \pm 3.362$ | $58.924 \pm 4.865$ | **<0.001** |
| $f$ (no units) | $0.30795 \pm 0.019$ | $0.33801 \pm 0.019$ | **<0.001** |
| $d$ (mm$^2$ s$^{-1}$) | $0.00147 \pm 7.5e^{-5}$ | $0.00159 \pm 8.4e^{-5}$ | **0.00167** |
| $d^*$ (mm$^2$ s$^{-1}$) | $0.01372 \pm 0.0006$ | $0.014106 \pm 0.0004$ | 0.0580 |
| $v$ (no units) | $0.273276 \pm 0.014$ | $0.298236 \pm 0.016$ | **<0.001** |
| $T_2^{f_b}$ (ms) | $112.6975 \pm 25.195$ | $129.1435 \pm 15.814$ | **0.00633** |
| $S_{O_2}$ (%) | $86.5019 \pm 7.186$ | $91.3252 \pm 6.078$ | **0.00234** |
| BG measurements | | | |
| Fetal arterial $S_{O_2}$ (%) | $50.7 \pm 15.4$ | $56.0 \pm 14.6$ | **0.0385** |
| Fetal arterial $P_{O_2}$ (mmHg) | $18.4 \pm 4.78$ | $20.4 \pm 4.74$ | **0.00693** |
| Fetal arterial $P_{CO_2}$ (mmHg) | $60.5 \pm 10.02$ | $62.9 \pm 11.6$ | **0.0121** |
| Fetal arterial pH | $7.283 \pm 0.0407$ | $7.276 \pm 0.0529$ | 0.294 |
| Maternal venous $S_{O_2}$ (%) | $86.8 \pm 5.58$ | $97.3 \pm 2.56$ | **<0.001** |
| Maternal venous $P_{O_2}$ (mmHg) | $64.5 \pm 11.5$ | $111 \pm 23.1$ | **<0.001** |
| Maternal venous $P_{CO_2}$ (mmHg) | $43.5 \pm 8.34$ | $46.7 \pm 9.23$ | 0.212 |
| Maternal venous pH | $7.376 \pm 0.0609$ | $7.347 \pm 0.0532$ | **0.0305** |

A paired sample $t$ test was performed to evaluate differences between maternal normoxia and hyperoxia states. Significant differences between the two groups are shown in bold ($P < 0.05$). Results are presented as the mean $\pm$ SD. The parameters presented are MRI placental blood relaxation time (T2), placental transverse relaxation time (T2*), feto-placental blood volume fraction ($f$), trophoblast apparent diffusivity ($d$), pseudo-diffusivity ($d^*$), maternal blood volume fraction ($v$), feto-placental blood relaxation time ($T_2^{f_b}$) and feto-placental oxygen saturation ($S_{O_2}$).

$T_2^{f_b}$ and 6% in fetal $S_{O_2}$. Maternal hyperoxia increased MRI-derived feto-placental relaxation time ($T_2^{f_b}$) by 14.6% and fetal $S_{O_2}$ in the DAo measured by BGA by 5.58% (Table 1). During maternal hyperoxygenation, feto-placental blood volume was significantly increased by 9.8% ($0.308 \pm 0.019$ *vs.* $0.338 \pm 0.019$, $P < 0.001$) and maternal blood volume fraction was significantly increased by 9.1% ($0.273 \pm 0.014$ *vs.* $0.298 \pm 0.016$, $P < 0.001$). Maternal hyperoxygenation also significantly increased the trophoblast apparent diffusivity by 8.2% ($0.00147 \pm 7.5e^{-5}$ mm$^2$ s$^{-1}$ *vs.* $0.00159 \pm 8.4e^{-5}$ mm$^2$ s$^{-1}$, $P = 0.00167$) but there was no significant difference in pseudo-diffusivity parameter.

## Relationships between fetal blood gas analysis and MRI

Linear regression analysis and correlation were performed for the MRI-derived parameters and BGA measurements (Figs 3 and 4). The overall regression analysis was statistically significant between the DECIDE MRI parameters and conventional BGA measures of fetal $P_{O_2}$ and $S_{O_2}$ (Fig. 3). MRI feto-placental $S_{O_2}$ significantly correlated with BGA-derived fetal arterial $S_{O_2}$ ($y = 0.191x - 133.7$, $R^2 = 0.780$, $r = 0.899$, $P < 0.001$)

(Fig. 3B) and fetal arterial $P_{O_2}$ ($y = 0.665x - 39.8$, $R^2 = 0.791$, $r = 0.889$, $P < 0.001$) (Fig. 3E). There was also a strong linear correlation between trophoblast diffusivity and BGA measurements of both fetal $S_{O_2}$ and $P_{O_2}$ (Fig. 3C and F). A significant linear relationship was observed between maternal venous measures and placental T2 and T2* (Fig. 4A). T2 significantly predicted maternal venous $P_{O_2}$ ($y = 2.97x - 312$, $R^2 = 0.873$, $r = 0.934$, $P < 0.001$) (Fig. 4B) and $S_{O_2}$ ($y = 0.985x + 37.8$, $R^2 = 0.758$, $r = 0.871$, $P < 0.001$) (Fig. 4C). A strong linear correlation was also observed between T2* and maternal venous $P_{O_2}$ ($y = 4.46x - 157.4$, $R^2 = 0.811$, $r = 0.901$, $P < 0.001$) (Fig. 4D) and $S_{O_2}$ ($y = 0.667x + 2.25$, $R^2 = 0.845$, $r = 0.919$, $P < 0.001$) (Fig. 4E).

The Bland–Altman plot in Fig. 5 compares the MRI feto-placental $S_{O_2}$ measurements against BGA measures of fetal oxygenation measurements from fetal descending aorta and shows good agreement between MRI feto-placental $S_{O_2}$ and gold-standard blood gas samples for both maternal normoxia and hyperoxia states. These differences in the MRI feto-placental $S_{O_2}$ result in a consistent bias of 35% for maternal normoxia and 34% for maternal hyperoxia. The 95% limits of agreement across all animals ranged from 21.6% to 49.4% for maternal normoxia and from 20.8% to 47.9 % for maternal hyperoxia.

### Changes in placental BOLD MRI with hyperoxygenation

The BOLD signal rose by 16–19% from maternal normoxia to hyperoxia. For all eight sheep in our study, the BOLD signal was significantly increased during maternal hyperoxia: $\Delta BOLD = 17.5 \pm 1.15\%$ (Fig. 1).

Figure 6 demonstrates a linear relationship between changes in MRI T2 and T2* and changes in BOLD signal intensities (Fig. 6*A* and *B*). Linear regression analysis between BOLD-derived and catheter-derived maternal venous $S_{O_2}$ showed a strong correlation and a good coefficient of determination ($r = 0.80$, $R^2 = 0.64$, $P = 0.017$) (Fig. 6*C*). No linear association was observed between BOLD-derived and maternal venous $P_{O_2}$ by BGA ($r = 0.54$, $R^2 = 0.29$, $P = 0.17$) (Fig. 6*D*).

### Changes in placental MRI parameters with placentome type

Table 2 shows the results of a two-way ANOVA to analyse the effect of maternal hyperoxia according to the placentome type. Simple main effects analysis shows that maternal hyperoxia had a statistically significant effect on all MRI-derived parameters except pseudo-diffusivity. In addition, simple main effect analysis shows that placentome type had a statistically significant effect on T2, T2*, feto-placental blood relaxation and trophoblast diffusivity MRI parameters ($P = 0.00197$, $<0.001$, $0.0234$

and $0.0193$). Two-way ANOVA revealed that there was no statistically significant interaction between the effects of maternal hyperoxia and placentome type.

Figure 7 shows the MRI parameter estimates for the two placentome types. There were significant differences between type A and type B placentomes for T2, T2* and *f* parameters. However, there was no evidence of a significant difference across type A and type B for the other MRI-derived parameters.

## Discussion

In this study of normal pregnant sheep, we have shown measurable changes in the placenta using multicontrast MRI before, during and after maternal hyperoxia and how changes on imaging relate to those from the gold-standard blood gas analyser. Our results show substantive changes on T2, T2* and diffusion-weighted MRI in both DECIDE and BOLD imaging with strong relationships between these methods. In addition, we have identified differences in the response to hyperoxia with placentome type including decreasing T2 and T2*.

Our results show that the DECIDE multicompartment model is sensitive to changes in maternal and feto-placental oxygenation. As a result of the DECIDE model that we used, the changes in compartment volume fraction reflect a change in the placental oxygen gradient. Increased oxygen in the deep placentome villi causes the estimated maternal blood volume fraction to increase

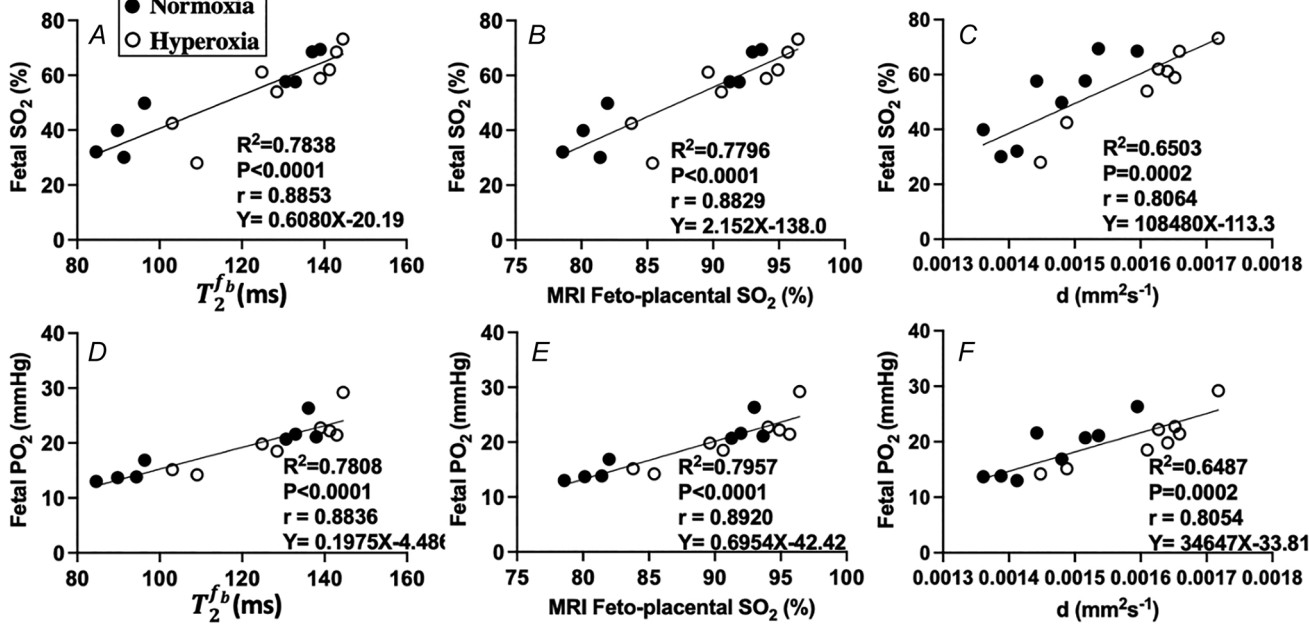

**Figure 3. Relationships between the fetal $S_{O_2}$ measured by a blood gas analyser (BGA) and the MRI-derived parameters**
Feto-placental blood relaxation time (*A*), $T_2^{fb}$, MRI feto-placental $S_{O_2}$ (*B*) and diffusivity, *d* (*C*). Relationships between the fetal $P_{O_2}$ measured by BGA and MRI parameters (*D–F*). Linear regression is depicted by solid black lines.

**Table 2. Average MRI parameters derived over all singleton pregnancies during maternal normoxia and hyperoxia for each placentome type**

| MRI parameter | Maternal normoxia (NX) | | Maternal hyperoxia (HYX) | | Effect of HYX ($P$) | Effect of type ($P$) | Interaction HYX and type ($P$) |
| --- | --- | --- | --- | --- | --- | --- | --- |
| | Type A ($n = 259$) | Type B ($n = 40$) | Type A ($n = 259$) | Type B ($n = 40$) | | | |
| T2 (ms) | $131.27 \pm 5.65$ | $124.13 \pm 3.67$ | $144.54 \pm 5.98$ | $139.22 \pm 2.78$ | <0.001 | 0.00197 | 0.619 |
| T2* (ms) | $53.06 \pm 3.69$ | $48.42 \pm 3.03$ | $60.33 \pm 4.19$ | $53.19 \pm 3.04$ | <0.001 | <0.001 | 0.386 |
| $f$ (no units) | $0.300 \pm 0.018$ | $0.308 \pm 0.017$ | $0.322 \pm 0.015$ | $0.348 \pm 0.019$ | <0.001 | 0.0234 | 0.213 |
| $d$ (mm$^2$s$^{-1}$) | $0.00149 \pm 7.7e^{-5}$ | $0.00144 \pm 8.5e^{-5}$ | $0.00164 \pm 9.6e^{-5}$ | $0.00154 \pm 8.20e^{-5}$ | <0.001 | 0.0193 | 0.551 |
| $d^*$ (mm$^2$ s$^{-1}$) | $0.0138 \pm 0.0007$ | $0.0136 \pm 0.0005$ | $0.0141 \pm 0.0003$ | $0.0139 \pm 0.0002$ | 0.117 | 0.372 | 0.755 |
| $v$ (no units) | $0.265 \pm 0.0197$ | $0.272 \pm 0.0136$ | $0.289 \pm 0.0162$ | $0.305 \pm 0.0191$ | <0.001 | 0.103 | 0.520 |
| $T_2^{f_b}$ (ms) | $114.35 \pm 23.004$ | $112.19 \pm 23.597$ | $130.51 \pm 14.184$ | $127.08 \pm 15.559$ | 0.0447 | 0.709 | 0.932 |
| $S_{O_2}$ (%) | $87.01 \pm 6.477$ | $86.43 \pm 6.720$ | $91.75 \pm 4.377$ | $90.62 \pm 4.519$ | 0.0449 | 0.689 | 0.898 |

Two-way ANOVA was performed to examine the effect of maternal hyperoxia (HYX) and placentome type on the MRI-derived parameters. Significant differences between placentome types in each oxygenation state are shown in bold ($P < 0.05$). Results are presented as the mean $\pm$ SD.

as the oxygen gradient along the materno-placental villi is reduced. Another interesting result is the increase in trophoblast diffusivity. This may indicate a more restricted cellular environment or a denser cellular space that therefore impedes oxygen transfer between maternal and fetal blood pools. Comparably, the estimated feto-placental blood volume fraction increases due to the increase in oxygenation of the feto-placental villi. Always vulnerable to the low signal-to-noise ratio, pseudo-diffusivity is often incapable of offering a statistically significant conclusion. Our results also showed a highly linear bias on a Bland-Altman plot, indicating that fetal oxygenation is overestimated by MRI at lower values compared to blood gas analysis. This is highly relevant because knowledge of this bias will allow future prediction and adaptation of the model in cases where lower oxygen saturation has clinical relevance. The results are comparable to those reported previously for

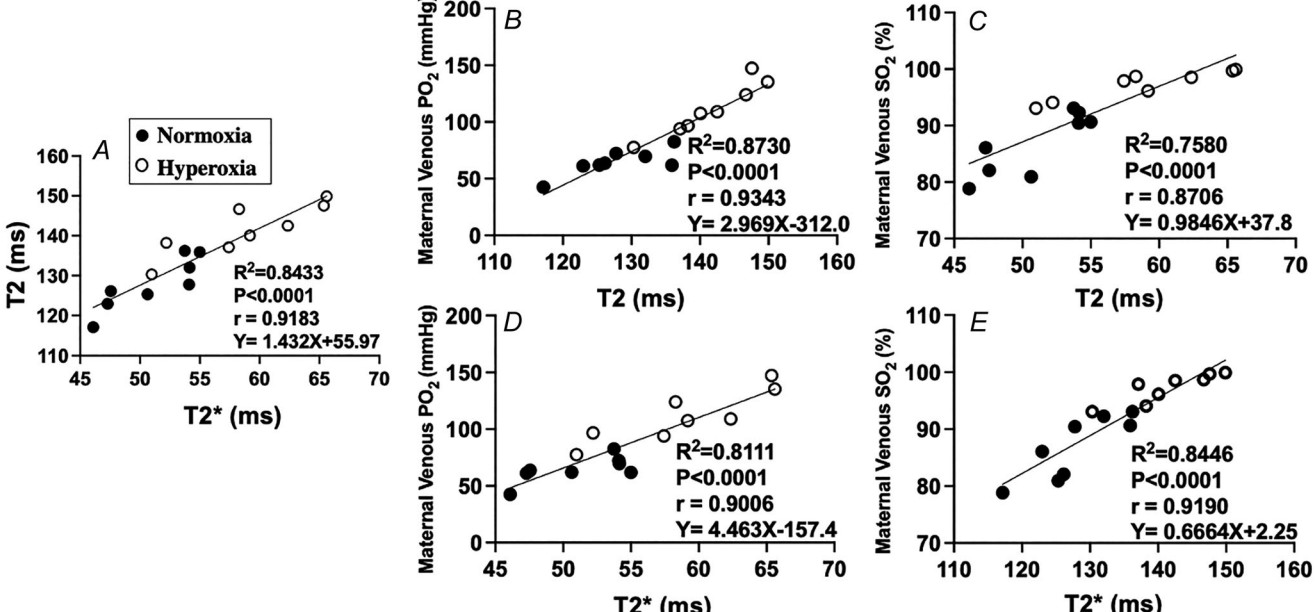

**Figure 4. Relationships between MRI placental parameters and blood gas measurements of $S_{O_2}$ and $P_{O_2}$**
*A*, Relationship between the observed MRI placental T2 and T2* parameters; and *B-E*, relationships between blood gas measurements of maternal oxygen saturation ($S_{O_2}$) and partial pressure of oxygen ($P_{O_2}$). Linear regression is depicted by solid black lines.

T2 MRI oximetry and fetal $S_{O_2}$ measurements by BGA in normal sheep subjects by Saini et al. (2020).

Previous MRI studies in sheep models of human pregnancy did not focus on placental oxygenation (Wedegärtner et al., 2006, 2010). This may be because the sheep placenta is different from the single discoid human placenta in that it consists of multiple placentomes (i.e. separate cotyledons), requiring segmentation of 40–80 regions of interest. In addition, the sheep placenta is functionally different in that it is assumed to be a concurrent haemodynamic system with more layers between maternal and fetal blood, which is less efficient than the counter-current system evidenced in the human placenta (McNanley & Wood, 2008; Schroder, 1995).

Quantitative analysis of changes in oxygenation is challenging because the BOLD signal is dependent on many biophysical and physiological parameters. To our knowledge, this is the first study in sheep placenta to correlate measurements of multimodal MRI techniques such as BOLD and DECIDE with directly measured blood gases during maternal hyperoxia. We analysed the BOLD effect induced by maternal hyper-oxygenation based on the signal intensities of placentome tissues obtained with multiple TE and with dynamic BOLD measurement at single TE. According to the BOLD concept, increased signal intensity corresponds to decreased deoxyhaemoglobin content and thus to increased tissue oxygenation. BOLD MRI demonstrated a significant increase in placentome oxygenation during maternal hyperoxia and this was observed on both

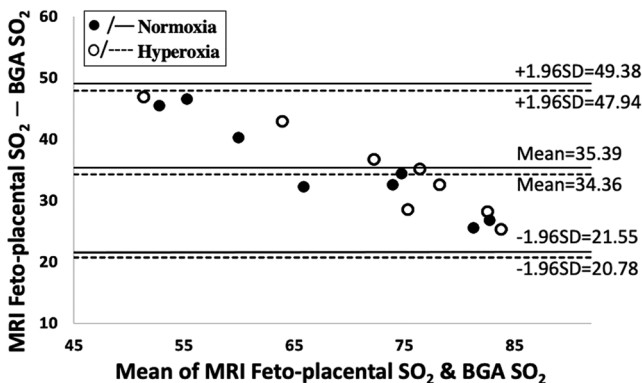

**Figure 5. Comparison of MRI feto-placental $S_{O_2}$ against BGA measured fetal descending aorta blood $S_{O_2}$ during the MRI scan**
Bland–Altman plot comparing MRI feto-placental oxygenation saturation ($S_{O_2}$) against reference blood gas analyser (BGA) measured fetal descending aorta blood $S_{O_2}$ sampled during the MRI scan. The y-axis shows the difference of fetal MRI estimates of oxygen saturation minus reference BGA $S_{O_2}$. Open circles and solid lines indicate the mean and the 95% confidence interval for the maternal normoxia. Closed circles and dashed lines indicate the mean and the 95% confidence interval respectively for the maternal hyperoxia.

multiple $T_E$ and single $T_E$ sequences (Sørensen et al., 2013 ; Wedegärtner et al., 2006). In the present study, we have shown how changes in dynamic BOLD MRI relate to quantitative changes in oxygen saturation measured by both MR relaxometry and by blood gas analysis. Future work could expand BOLD MRI in human placentas to identify differences in oxygen uptake in different parts of the placenta by drawing smaller regions of interest across the placenta, as well as the total placental oxygenation by drawing a region of interest covering the entire placenta. The regional placental oxygenation in the human placenta may help us to reflect the differences in oxygen uptake that we observed in different placentome types. In addition, the perfusion and oxygenation properties of different placentome types in the sheep is an area of future further study. It is known that placentomes progressively change type during gestation from A/B to C/D and it appears that this process is accelerated after carunclectomy (Flouri et al., 2021). The driving process for change may well be related to oxygen saturation and the response of placentomes to maternal hyper-oxygenation may differ between placentome type because of oxygen concentration differences caused by the relative volumes of fetal and maternal tissue within a placentome.

Maternal hyperoxia increases fetal oxygen supply (Battaglia et al., 1968). This has been demonstrated in the human fetus using cordocentesis (Nicolaides et al., 1987), pulse oximetry (Haydon et al., 2006) and postpartum umbilical cord blood sampling (Ngan et al., 2002). Using the BOLD MRI technique in the sheep fetus, we have demonstrated that maternal hyperoxygenation increases fetal blood oxygenation. This effect is probably mediated by increased $P_{O_2}$ in the maternal blood of the intervillous space, which improves the diffusion of oxygen across a higher gradient in the placenta from the mother to the fetus (Carter, 1999). Although we did not observe a systematic change in fetal heart rate during BOLD, changes in umbilical heart rate flow could provide a fetal compensatory mechanism for hyperoxia or normoxia. The addition of imaging such as phase contrast flow measurement could help disentangle dynamic changes in placental flow and perfusion further.

One limitation of the present study was the small number of animals. However, our data were powerful enough to demonstrate significant trends as a result of maternal hyperoxia. It is possible that, with higher numbers, additional MR parameters will show significant variation with hyperoxygenation. Our study comprises data from one time point in late gestation. Furthermore, the experimental setting with surgery and anaesthesia to ensure the welfare of the ewe may have affected our results. General anaesthesia with artificial ventilation and the ewe in the lateral position may affect maternal and fetal oxygen exchange in the placenta and oxygen distribution in the fetus to various degrees. However, it must be noted

that general anaesthesia is not required for MRI studies and pregnant women are generally willing to lie still in an MRI for sessions of 60–90 min (Flouri et al., 2022; Melbourne et al., 2019; Saini et al., 2021; Saini et al., 2020). One of the drawbacks of BOLD technique is the limited spatial resolution. Blood oxygenation varies by region and, consequently, the image signal is very low compared to statistical noise. A general problem in fetal imaging is fetal motion. In the present study, there were no severe motion artefacts as a result of general anaesthesia of the mother, which sedates the fetus as well. A further limitation of the study was not including maternal hypoxia, which would

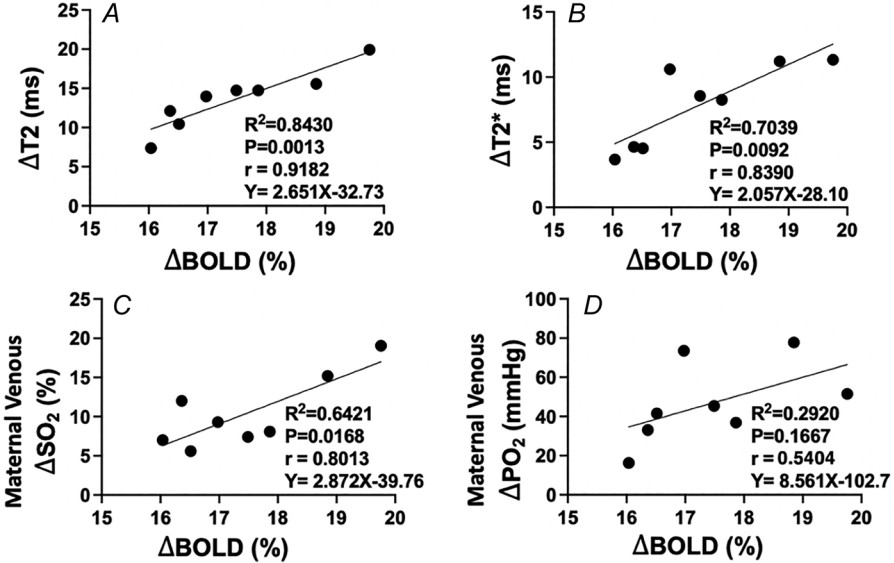

**Figure 6. Changes of BOLD signal**
Scatter plots show comparison of placentome change in BOLD ($\Delta$ BOLD); *A*, with the change in T2 ($\Delta$T2) and *B*, T2* ($\Delta$ T2*) MRI-derived parameters, as well as changes in blood gas measurements of *C*, maternal oxygen saturation ($\Delta S_{O_2}$) and *D*, partial pressure of oxygen ($\Delta P_{O_2}$). Linear regression is depicted by solid black lines.

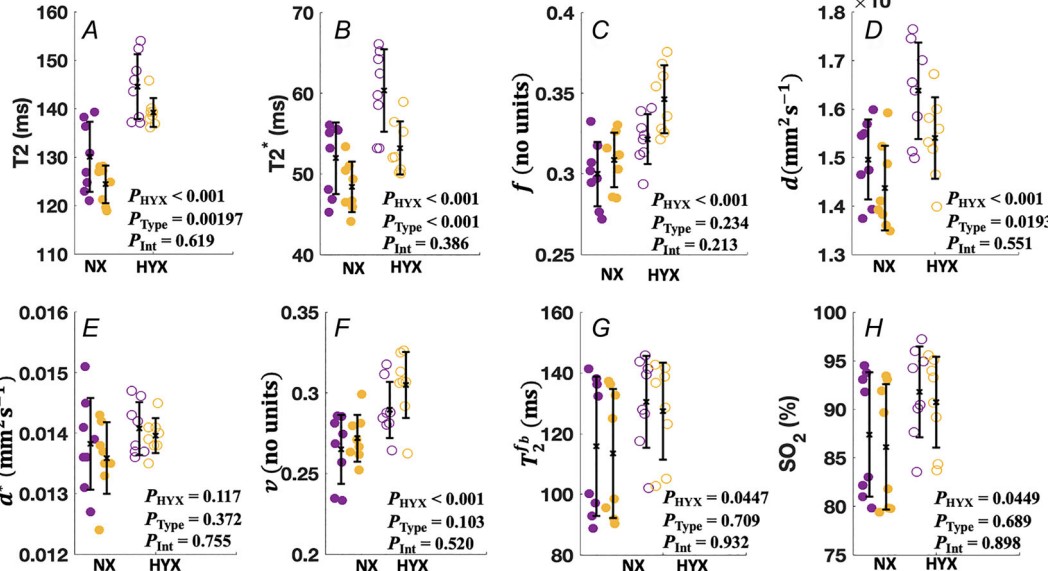

**Figure 7. Boxplots illustrate the changes in MRI parameters during maternal normoxia (NX) and maternal hyperoxia (HYX) for each placentome type in all singleton pregnancies**
*A-H*, Each plot shows the median as well as the 25th and 75th percentile. Two-way ANOVA was performed to examine the effect of maternal hyperoxia and placentome type on the MRI-derived parameters. $P_{HYX}$; effect of maternal hyperoxia; $P_{Type}$; effect of placentome type; $P_{Int}$; interaction effect of maternal hyperoxia and placentome type. Purple, type A placentomes; yellow, type B placentomes; closed symbols, normoxia; open symbols, hyperoxia.

enable a larger range of oxygen levels to be measured and a wider correlation obtained with human studies (Hutter et al. 2010). However, maternal hypoxia was not included in the present study because the MRI session duration required to perform measurements in three oxygenation states would have been longer than the pregnant sheep could tolerate. Future studies will investigate changes in placental oxygenation during induced maternal hypoxia using BOLD MRI. Another limitation of the current DECIDE model is that it does not currently estimate an individual haematocrit (Hct) for use in the oxygen saturation model. In the case of some pathologies such as fetal growth restriction, differences in Hct may cause a slight bias, although this is expected to be small as described in Jani et al. (2023). However, in studies of acute changes in oxygenation status, the estimation of oxygen difference is less affected by the baseline $S_{O_2}$.

We have shown the sensitivity of new methods of measuring placental function to known changes in fetal and maternal oxygenation. Being able to disentangle these changes is critical towards developing clinically relevant techniques for the assessment of placental pathology. The clinical perspective of these results may be that DECIDE-like and BOLD MRI techniques provide non-invasive techniques in the difficult field of assessing placental function. In particular, this method may be useful for exploring differences in placental oxygenation and function at the level of the cotyledon, providing more information on variation in structure and function within the placenta. The adaptation of these methods for the differential diagnosis of placental insufficiency in pathological pregnancies, such as in fetal growth restriction, represents a tangible future clinical pathway.

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

## Additional information

### Data availability statement

All data supporting the results are presented in the published article. MRI images can be accessed upon reasonable request to the senior authors.

### Competing interests

The authors declare that they have no competing interests.

### Author contributions

A.D., J.L.M. and A.M. were responsible for conception and design of the study. D.F., J.R.T.D., S.L.H., G.W., A.D., J.L.M. and A.M. were involved in acquisition, analysis and interpretation of data for the work. D.F., J.L.M. and A.M. drafted the article. All authors contributed to and have read and approved the final version of the manuscript submitted for publication. All authors agree to be accountable for all aspects of the work ensuring that questions related to the accuracy or integrity of any part of the work are appropriately investigated and resolved. All persons designated as authors qualify for authorship, and all those who qualify for authorship are listed.

### Funding

D.F. was supported by Marie Sklodowska-Curie Post-doctoral Fellowship (101108945 – InSilicoPlacenta). The animal work was supported by the Australian Research Council (DP190102263 and DP220103289) and BBSRC (BB/Y514214/1). AM was supported by BBSRC (BB/Y514214/1) the MRC (MR/X010007/1) and the NIH (R01 HD108833). D.F., A.L.D. and A.M. were also supported by EPSRC (NS/A000027/1).

### Acknowledgements

We acknowledge the technical assistance of the National Imaging Facility, a NCRIS capability, at PIRL, SAHMRI, as well as support in animal care by members of the Early Origins of Adult Health Research Group.

Open access publishing facilitated by University of South Australia, as part of the Wiley - University of South Australia agreement via the Council of Australian University Librarians.

### Keywords

DECIDE, maternal hyperoxia, magnetic resonance imaging, oxygen, placenta, sheep

## Supporting information

Additional supporting information can be found online in the Supporting Information section at the end of the HTML view of the article. Supporting information files available:

**Peer Review History**

