## [Peer Review History · The Journal of Physiology]

Feasibility of multi-modal magnetic resonance imaging to assess maternal hyperoxygenation in sheep pregnancy

Dimitra Flouri, Jack RT Darby, Stacey L Holman, Georgia K Williams, Vasileios Vavourakis, Anna Louise David, Janna L Morrison, and Andrew Melbourne

DOI: 10.1113/JP287272

Corresponding author(s): Dimitra Flouri (flouri.dimitra@ucy.ac.cy)

Review Timeline:

Submission Date:	30-Jul-2024
Editorial Decision:	24-Oct-2024
Revision Received:	05-Dec-2024
Editorial Decision:	16-Jan-2025
Revision Received:	17-Jan-2025
Accepted:	21-Jan-2025

Senior Editor: Kim Barrett

Reviewing Editor: Laura Bennet

Transaction Report:

Dear Dr Flouri,

Re: JP-TFP-2024-287272 "Feasibility of multi-modal magnetic resonance imaging to assess maternal hyperoxygenation in sheep pregnancy" by Dimitra Flouri, Jack RT Darby, Stacey L Holman, Georgia K Williams, Vasileios Vavourakis, Anna Louise David, Janna L Morrison, and Andrew Melbourne

Thank you for submitting your manuscript to The Journal of Physiology. It has been assessed by a Reviewing Editor and by 2 expert referees and we are pleased to tell you that it is acceptable for publication following satisfactory revision.

REVISION CHECKLIST:

Please upload two versions of your manuscript text: one with all relevant changes highlighted and one clean version with no changes tracked. The manuscript file should include all tables and figure legends, but each figure/graph should be uploaded as separate, high-resolution files. The journal is now integrated with Wiley's Image Checking service. For further details, see: <https://www.wiley.com/en-us/network/publishing/research-publishing/trending-stories/upholding-image-integrity-wileys->

image-screening-service

We look forward to receiving your revised submission.

Yours sincerely,

Kim Barrett
Senior Editor
The Journal of Physiology

REQUIRED ITEMS:

- Author photo and profile. First or joint first authors are asked to provide a short biography (no more than 100 words for one author or 150 words in total for joint first authors) and a portrait photograph. These should be uploaded and clearly labelled together in a Word document with the revised version of the manuscript. See Information for Authors for further details.
 - Please upload separate high-quality figure files via the submission form.
 - Please ensure that the Article File you upload is a Word file.
 - Please include an Abstract Figure file, as well as the Figure Legend text within the main article file. The Abstract Figure is a piece of artwork designed to give readers an immediate understanding of the research and should summarise the main conclusions. If possible, the image should be easily 'readable' from left to right or top to bottom. It should show the physiological relevance of the manuscript so readers can assess the importance and content of its findings. Abstract Figures should not merely recapitulate other figures in the manuscript. Please try to keep the diagram as simple as possible and without superfluous information that may distract from the main conclusion(s). Abstract Figures must be provided by authors no later than the revised manuscript stage and should be uploaded as a separate file during online submission labelled as File Type 'Abstract Figure'. Please also ensure that you include the figure legend in the main article file. All Abstract Figures should be created using BioRender. Authors should use The Journal's premium BioRender account to export high-resolution images. Details on how to use and access the premium account are included as part of this email.
-

Reviewing Editor:

Methods Details:

The authors need to provide information on and when and how the ewes and their fetuses were euthanised. I imagine they were euthanised at the end of the procedures, and so analgesia is not needed, but I cannot see this in text.

Ethics Concerns:

I have no specific concerns, it is just a matter of providing some missing information.

Comments to the Authors:

Thank you for your submission. The reviewers have provided areas they would like to see addressed in greater detail. In particular both reviewers note that there are already studies in sheep, rats and humans (Sorensen and colleagues) and thus there is a need to further emphasise the key new information and the importance of this information with regards to translation to humans. Reviewer 2 suggests a translational perspective section in your discussion may be one approach to address this. For our readers, it is important to discuss the differences between human and sheep placentae with regards to your results and please provide details about how and when the sheep were euthanised

Referee #2:

The manuscript seeks to validate the feasibility of the DECIDE MRI framework for evaluating changes in fetoplacental oxygenation, especially its sensitivity to detect the higher end of this range after maternal hyperoxygenation treatment. Overall, the well-organized manuscript clearly shows that the DECIDE framework corresponds well to direct blood gas and BOLD MRI measurements.

Primary Concern: The most pressing concern with this manuscript is the novelty factor. This group has published several articles before using the DECIDE framework in a similar sheep model of pregnancy (PMID: 38606906 & 36031385). In both studies, this group has already shown the capability of this technology to assess oxygenation in the ranges seen in the present study (i.e. T2fb between 100-150ms and MRI fetoplacental SO₂ between 85-90%). Thus, it is unclear why further validation with maternal hyperoxia is necessary, especially given that the ranges in the present study overlap significantly with their past work.

Minor Concerns:

1) There is a significant emphasis on the capability of this technology to replace cordocentesis; however, the current DECIDE framework cannot quantify fetal Hb concentration. It is known that the fetus can initiate several compensatory adaptations to hypoxemia, such as increasing Hb concentration. Thus, without a direct measure of fetal Hb, MRI-derived parameters may overestimate the true extent of hypoxemia. This should be addressed as a limitation of the clinical translatability of this technology.

2) From Figure 1, placentome types appear to be visually similar. Did you consider assessing interobserver variation for placentome classification?

3) Figure 1 seems out of place, especially since it is only referenced in the latter half of the results section.

- 4) In Figure 2, the normoxia and hyperoxia MRI measurements are not always within the exact same imaging plane. Could this have affected measurements?
 - 5) I recommend avoiding using MRI-derived variables (d , d^* , v etc.) in the main text to make reader comprehension easier, especially for those unfamiliar with this technology.
 - 6) In Figure 3, if MRI fetoplacental SO_2 (%) is derived from T2fb, is it necessary to show both linear regressions since they are virtually identical? Additionally, why are the R^2 and r values for the regression different between A-B and D-E? If SO_2 is derived from T2 and all other variables are held constant, as stated in equation 2, shouldn't they be the same?
 - 7) In Figure 3, there are some inconsistencies between the R^2 and r values for some regressions. Consider recalculating.
 - 8) In Figure 6D, the R^2 value is identical to the r value.
 - 9) Minor grammatical errors and confusing sentence structures throughout. Please proofread.
-

Referee #3:

This is an important study that assesses the utility of non-invasive magnetic resonance imaging to assess foetal oxygenation status in pregnant sheep exposed to hyperoxia. The rationale for the study has been clearly identified - the need to assess foetal oxygen levels without recourse to intramniotic sampling of the umbilical cord blood - and the methods are sound. However, it is a lost opportunity that reductions in maternal inhaled oxygen were not also assessed, given the clinical significance of reductions in foetal oxygen.

When referring to Fig. 1, please describe the cyclical fluctuations in BOLD signal: are these related to foetal movements and, if so, how are they dealt with?

It would be worth defining the four different types of placentomes, given that only types A and B are included in this study.

On line 128, the authors refer to their use of "advanced MRI" techniques, but these are really standard tools for brain imaging.

On line 479, the authors state that "maternal hyperoxia increases fetal oxygen supply." Is this associated with an increase or decrease in umbilical blood flow and, if so, how is this dealt with in the DECIDE model and when considering the aim of quantifying the BOLD signal.

Have the authors considered pseudo-continuous arterial spin labelling (pcASL) to measure umbilical blood flow?

The Discussion section is remarkably short. The authors should discuss the differences between human and ovine pregnancies, particularly with respect to the existence of placentomes. Moreover, the Discussion would benefit from consideration of how MRI could identify differences in oxygen uptake in different parts of the human placenta; presumably, the differential changes in oxygen uptake in the type A and B placentomes is relevant here?

There should also be discussion on the significance of reduced foetal blood flow, and a comment that not including hypoxia in the experimental design is a limitation. The ultimate clinical applicability of this approach should be elaborated on with a Translational Perspective, including discussion on how foetal motion would be corrected if this approach is extended to humans. The work of Sorensen, cited by the authors, should be expanded on and the advantages of the experimental approach used here (correlation of BOLD signal intensity, etc, with directly measured blood gases) emphasised.

END OF COMMENTS

We are grateful to both reviewers for their helpful comments and suggestions. Detailed responses to each of the editor's and reviewer's comments are below.

Comments to the Authors:

Thank you for your submission. The reviewers have provided areas they would like to see addressed in greater detail. In particular both reviewers note that there are already studies in sheep, rats and humans (Sorensen and colleagues) and thus there is a need to further emphasise the key new information and the importance of this information with regards to translation to humans. Reviewer 2 suggests a translational perspective section in your discussion may be one approach to address this. For our readers, it is important to discuss the differences between human and sheep placentae with regards to your results and please provide details about how and when the sheep were euthanised.

We thank the panel for raising these points. We have added details about how the sheep were euthanised in the methods section.

Line 169–171: *“At 124-125 days gestation, ewes were humanely killed with an overdose of sodium pentobarbitone (Randlab Australia Pty Ltd, Revesby, New South Wales, Australia) The fetus was delivered via hysterotomy and weighed before tissues were collected for use in subsequent studies.”*

As the Editor suggested, we have added the following text in the discussion regarding the differences between the human and sheep placentae.

Lines 400–405: *“Previous MRI studies in sheep models of human pregnancy did not focus on placental oxygenation (Wedegärtner et al. 2006, Wedegärtner et al. 2010). This may be because the sheep placenta is different from the single discoid human placenta in that it consists of multiple placentomes (i.e., separate cotyledons), requiring segmentation of 40-80 regions of interest. In addition, the sheep placenta is functionally different in that it is considered a concurrent hemodynamic system, which is less efficient than the counter current system in the human placenta (Schroder et al. 1995).”*

Referee #2:

The manuscript seeks to validate the feasibility of the DECIDE MRI framework for evaluating changes in fetoplacental oxygenation, especially its sensitivity to detect the higher end of this range after maternal hyperoxygenation treatment. Overall, the well-organized manuscript clearly shows that the DECIDE framework corresponds well to direct blood gas and BOLD MRI measurements.

1. **Primary Concern:** The most pressing concern with this manuscript is the novelty factor. This group has published several articles before using the DECIDE framework in a similar sheep model of pregnancy (PMID: 38606906 & 36031385). In both studies, this group has already shown the capability of this technology to assess oxygenation in the ranges seen in the present study (i.e., T2fb between 100-150ms and MRI fetoplacental SO2 between 85-90%). Thus, it is unclear why further validation with maternal hyperoxia is necessary, especially given that the ranges in the present study overlap significantly with their past work.

R2.1. – We thank the reviewer for this comment. We have previously performed similar studies measuring SO_2 in human and sheep using the DECIDE model but without changing the maternal oxygenation status. Maternal hyperoxia is known to increase fetal oxygen supply provided there is normal placental function. This has been demonstrated in the human fetus by cordocentesis, pulse oximetry and postpartum umbilical cord blood sampling. Although the DECIDE framework has been used in previous studies to assess long-term oxygenation trends with gestation, its sensitivity to acute maternal oxygen administration has not yet been investigated. Moreover, a novel aspect of the present study was the use of both T2^* and BOLD MRI to demonstrate that these modalities are capable of detecting maternal hyperoxia induced increases fetal tissue oxygenation. The advantage of using DECIDE in this framework is the addition of a multi-compartment physiological model that links signal change more strongly to physiological changes in oxygenation and compares this to two imaging modalities (T2^* and BOLD) that do not quantitatively assess oxygen saturation. Thus, here we further validated how results of fetal oxygenation from the BOLD data related to those derived from the DECIDE model and furthermore the correlation of the BOLD signal with directly measured blood gases.

We have added the following text in the discussion to emphasize the novelty of our study.

Lines 408–410: *“To our knowledge, this is the first study in sheep placenta to correlate measurements of multi-modal MRI techniques such as BOLD and DECIDE with directly measured blood gases during maternal hyperoxia.”*

2. There is a significant emphasis on the capability of this technology to replace cordocentesis; however, the current DECIDE framework cannot quantify fetal Hb concentration. It is known that the fetus can initiate several compensatory adaptations to hypoxemia, such as increasing Hb concentration. Thus, without a direct measure of fetal Hb, MRI-derived parameters may overestimate the true extent of hypoxemia. This should be addressed as a limitation of the clinical translatability of this technology.

R2.2. – We agree that this is a relevant limitation. The proposed model does allow for the effect of Hct to be modelled, but this is set to a fixed value, appropriate for the acute study here, and in future this could perhaps be estimated by a comparable T1 quantification technique (although the errors in this measurement are likely to be large) or more finely chosen population/gestational averages. However, it is relevant that the dominant effect on T2 is the oxygen saturation, which is about an order of magnitude larger than that due to Hct differences. It is indeed known that for conditions such as FGR, Hb/Hct increases by a small amount, and this would be relevant [AM4] in a future study of a pathological condition as opposed to the paired study shown here during which Hct remains constant.

We have added the following text in the discussion.

Line 462–466: *“Another limitation of the current DECIDE model is that it does not currently estimate an individual hematocrit (Hct) for use in the oxygen saturation model. In the case of some pathologies such as FGR, differences in Hct may cause a slight bias (FGR may cause a small 2g/dl increase in haemoglobin, corresponding to <1% change in measured FO_2 as described in (Jani et al. 2023)). However, in studies*

of acute changes in oxygenation status, the estimation of oxygen difference is less affected by the baseline SO₂.”

3. From Figure 1, placentome types appear to be visually similar. Did you consider assessing interobserver variation for placentome classification?

R2.3. – We thank the reviewer for pointing this. It is true that there is a chance for misclassification. We have previously (*Flouri et al. 2019 Placenta*) investigated the interrater variability for placentome classification by two raters blinded to ewe information. Rater 1 has extensive experience working with MRI data, whereas Rater 2 is an expert in the examination and classification of sheep placentomes when tissue is collected at post-mortem. The interrater test was performed on 122 placentomes. The overall agreement between the two raters was very good ~88% (107 placentomes).

We have also added the following text in the methods.

Line 282–284: “*Placental MRI was used to morphologically classify placentomes into two types (A & B) using the classification system defined in a previous study (Flouri et al. 2021) in which we investigated the reproducibility of placentome classification between different types.*”

4. Figure 1 seems out of place, especially since it is only referenced in the latter half of the results section.

R2.4. – We appreciate the comment; however, we feel that Figure 1 portrays the BOLD effect during normoxia and hyperoxia states. Figure 1 was also used to add more details about the types of placentomes as requested by Reviewer 3 (R3.2.). We have added the following text in the methods to support this.

Line 260–262: “*The BOLD signal of each placentome region of interest (ROI) was recorded during the entire 10- min BOLD scan and for each ROI the BOLD signal at each time point was normalized using the mean BOLD signal of the initial 2-min of normoxia as a reference (Figure 1).*”

Line 286–291: “*Placentomes concave in shape with the maternal tissue (lighter grey-coloured) surrounding the fetal tissue (black) were classified as type A placentomes (Figure 1A). Type B and C placentomes are intermediate in shape. Type B placentomes consist of fetal tissue beginning to grow over the surrounding maternal tissue where type C placentomes consists of a larger portion of fetal tissue that has begun to surround maternal tissue. Type D placentomes contain mostly fetal tissue which surrounds the maternal tissue (Flouri et al. 2021).*”

5. In Figure 2, the normoxia and hyperoxia MRI measurements are not always within the exact same imaging plane. Could this have affected measurements?

R2.5. – We agree with the reviewer that the MRI measurements are not always within the exact same imaging plane. In this study we performed a 3D approach that segments a 3D volume of the image (i.e., draw region of interest in a volume-by-volume manner) unlike 2D methods that segment only one slice of the image. 3D segmentation is more robust to small changes caused by maternal breathing motion or fetal movements (*Flouri et al. 2019*).

6. I recommend avoiding using MRI-derived variables (d , d^* , v etc.) in the main text to make reader comprehension easier, especially for those unfamiliar with this technology.

R2.6. – We have removed the abbreviations of MRI-derived parameters in the main text and replaced them with consistent text descriptions.

7. In Figure 3, if MRI fetoplacental SO₂ (%) is derived from T₂^{fb}, is it necessary to show both linear regressions since they are virtually identical? Additionally, why are the R² and r values for the regression different between A-B and D-E? If SO₂ is derived from T₂ and all other variables are held constant, as stated in equation 2, shouldn't they be the same?

R2.7. – To calculate fetoplacental SO₂ for known T₂ we fitted the empirical curve Equation (2). Equation (2) is a nonlinear function, so the conversion of SO₂ from known T₂ is not guaranteed to be the same as T₂. Thus, we showed both regressions, but we do appreciate that the values shown are across a relatively linear range of this function.

8. In Figure 3, there are some inconsistencies between the R² and r values for some regressions. Consider recalculating.

R2.8. – We thank the reviewer for pointing this. We have carefully calculated the R² and r for the regressions and made the proper amendments in Figure 3 legends.

Figure 3: Relationships between the fetal SO₂ measured by a blood gas analyser (BGA) and the MRI-derived parameters, (A) fetoplacental blood relaxation time, T₂^{fb}, (B) MRI fetoplacental SO₂ and (C) diffusivity, d . Relationships between the fetal PO₂ measured by BGA and MRI parameters (D-F). Linear regression is depicted in solid black lines.

9. In Figure 6D, the R² value is identical to the r value.

R2.9. – We thank the reviewer spotting this typo. We corrected the value of ‘ r ’ accordingly.

10. Minor grammatical errors and confusing sentence structures throughout. Please proofread.
 R2.10. – We have proofread the manuscript and corrected minor grammatical errors where found.

Referee #3:

This is an important study that assesses the utility of non-invasive magnetic resonance imaging to assess foetal oxygenation status in pregnant sheep exposed to hyperoxia. The rationale for the study has been clearly identified - the need to assess foetal oxygen levels without recourse to intramniotic sampling of the umbilical cord blood - and the methods are sound. However, it is a lost opportunity that reductions in maternal inhaled oxygen were not also assessed, given the clinical significance of reductions in foetal oxygen.

We greatly appreciate the reviewer for their positive feedback on our work.

- When referring to Fig. 1, please describe the cyclical fluctuations in BOLD signal: are these related to fetal movements and, if so, how are they dealt with?
 R3.1. – The cyclical fluctuations occurred in some examples including this one and we were not able to reproduce the source. The period of the signal fluctuations is very long (about 2mins) suggesting that they are probably not related to fetal or maternal movements and were presumably electro-mechanical in nature. To account for these, we did experiment with fitting a coupled sinusoid alongside the sigmoid function. We were able to closely approximate the cyclic fluctuations, but this did not impact the fitted parameters of the sigmoid. In the interests of simplicity, we did not include this analysis in the manuscript.
- It would be worth defining the four different types of placentomes, given that only types A and B are included in this study.
 R3.2. – We have added more details in the methods as suggested.
Line 286–291: “Placentomes concave in shape with the maternal tissue (lighter grey-coloured) surrounding the fetal tissue (black) were classified as type A placentomes (Figure 1A). Type B and C placentomes are intermediate in shape. Type B placentomes

consist of fetal tissue beginning to grow over the surrounding maternal tissue where type C placentomes consists of a larger portion of fetal tissue that has begun to surround maternal tissue. Type D placentomes contain mostly fetal tissue which surrounds the maternal tissue (Flouri et al. 2021)."

3. On line 128, the authors refer to their use of "advanced MRI" techniques, but these are really standard tools for brain imaging.

R3.3. – Indeed, these techniques are standard in brain imaging; however, use of the ‘advanced MRI’ term is relative and in abdominal imaging adoption of such sequences is less widespread as there are additional technical challenges compared to brain imaging. However, we are happy to revise if deemed necessary.

4. On line 479, the authors state that "maternal hyperoxia increases fetal oxygen supply." Is this associated with an increase or decrease in umbilical blood flow and, if so, how is this dealt with in the DECIDE model and when considering the aim of quantifying the BOLD signal.

R3.4. – IVIM and DECIDE modelling is relatively insensitive to flow velocity, identifying perfusing blood rather than the net speeds, so this is a relevant comment. In a future work, we will interrogate further at the effect on fetal blood flows. As a surrogate though, this may be correlated with changes in fetal heart rate which we did not observe in this study. Also, our other studies of fetal blood flow with maternal hyperoxygenation suggest that there is no change in blood flow with UV [Unpublished].

We have added the following to explain this in the revised manuscript:

Line 437–441: “Although we did not observe a change in fetal heart rate during BOLD, changes in umbilical flow could provide a fetal compensatory mechanism for hyperoxia or normoxia. The addition of imaging such as a phase contrast flow measurement could help disentangle dynamic changes in placental flow and perfusion further.”

5. Have the authors considered pseudo-continuous arterial spin labelling (pcASL) to measure umbilical blood flow?

R3.5. – Thank you for this suggestion. It is indeed possible to measure umbilical blood flow with ASL, but this measure is exceptionally prone to noise and has not been validated. Phase contrast imaging may also provide a method for this from either a section of perpendicular cord, or from the fetus at the umbilical cord entry point. A full study would likely need to make use of an alternative technology such as ultrasound to be confident of the MRI results.

6. The Discussion section is remarkably short. The authors should discuss the differences between human and ovine pregnancies, particularly with respect to the existence of placentomes. Moreover, the Discussion would benefit from consideration of how MRI could identify differences in oxygen uptake in different parts of the human placenta; presumably, the differential changes in oxygen uptake in the type A and B placentomes is relevant here?

R3.6. – We have added the following text in the discussion.

Lines 400–405: “Previous MRI studies in sheep models did not focus on placental oxygenation (Wedegärtner et al. 2006, Wedegärtner et al. 2010). This may be because the sheep placenta is different from the single discoid human placenta in that it consists of multiple placentomes (i.e., separate cotyledons), requiring segmentation of 40-80

regions of interest. In addition, the placenta sheep is considered a concurrent hemodynamic system, which is less efficient than the counter current system in the human placenta (Schroder et al. 1995)."

Lines 418–429: *"Future work could expand BOLD MRI in human placentas to identify differences in oxygen uptake in different parts of the placenta by drawing smaller regions of interest across the placenta, as well as the total placental oxygenation by drawing a region of interest covering the entire placenta. The regional placental oxygenation in the human placenta may reflect the differences in oxygen uptake that we observed in different placentome types. In addition, the perfusion and oxygenation properties of different placentome types in the sheep is an area of future further study. It is known that placentomes progressively change type during gestation from A/B to C/D and it appears that this process is accelerated after carunclectomy (Flouri et al. 2021). The driving process for this change may well be related to oxygen saturation and the response of placentomes to maternal hyperoxygenation may differ between placentome type due to oxygen concentration differences caused by the relative volumes of fetal and maternal tissue within a placentome."*

7. There should also be discussion on the significance of reduced fetal blood flow, and a comment that not including hypoxia in the experimental design is a limitation. The ultimate clinical applicability of this approach should be elaborated on with a Translational Perspective, including discussion on how foetal motion would be corrected if this approach is extended to humans. The work of Sorensen, cited by the authors, should be expanded on and the advantages of the experimental approach used here (correlation of BOLD signal intensity, etc, with directly measured blood gases) emphasised.

R3.7. – We agree with the reviewer that not including hypoxia in the experimental design is a limitation of the study. However, it should be noted that the duration of the MRI session would have been longer than the pregnant sheep could tolerate due to the requirement for anesthesia in sheep. We have added the following text in the discussion.

Lines 456–462: *"A further limitation of the study was not including maternal hypoxia in the study, which would enable a larger range of oxygen levels to be measured and a wider correlation obtained with human studies (Hutter et al. 2010). However, maternal hypoxia was not included in the study since the MRI session duration to perform measurements in 3 oxygenation states would have been longer than the pregnant sheep could tolerate. Future studies will investigate changes in placental oxygenation during induced maternal hypoxia using BOLD MRI."*

We also added the following text in the discussion to emphasise the novelty of our work.

Lines 408–410: *"To our knowledge, this is the first study in sheep placenta to correlate measurements of multi-modal MRI techniques such as BOLD and DECIDE with directly measured blood gases during maternal hyperoxia."*

Reviewing Editor:

Methods Details:

The authors need to provide information on and when and how the ewes and their fetuses were euthanised. I imagine they were euthanised at the end of the procedures, and so analgesia is not needed, but I cannot see this in text.

We thank the editor for this important comment. We have added the following text in the methods section.

Line 169–171: *“At 124-125 days gestation, ewes were humanely killed with an overdose of sodium pentobarbitone (Randlab Australia Pty Ltd, Revesby, New South Wales, Australia) The fetus was delivered via hysterotomy and weighed before tissues were collected for use in subsequent studies.”*

Ethics Concerns:

I have no specific concerns; it is just a matter of providing some missing information.

Dear Dr Flouri,

Re: JP-TFP-2024-287272R1 "Feasibility of multi-modal magnetic resonance imaging to assess maternal hyperoxygenation in sheep pregnancy" by Dimitra Flouri, Jack RT Darby, Stacey L Holman, Georgia K Williams, Vasileios Vavourakis, Anna Louise David, Janna L Morrison, and Andrew Melbourne

Thank you for submitting your manuscript to The Journal of Physiology. It has been assessed by a Reviewing Editor and by 2 expert referees and we are pleased to tell you that it is acceptable for publication following satisfactory revision.

REVISION CHECKLIST:

Please upload two versions of your manuscript text: one with all relevant changes highlighted and one clean version with no changes tracked. The manuscript file should include all tables and figure legends, but each figure/graph should be uploaded as separate, high-resolution files. The journal is now integrated with Wiley's Image Checking service. For further details, see: <https://www.wiley.com/en-us/network/publishing/research-publishing/trending-stories/upholding-image-integrity-wileys->

image-screening-service

We look forward to receiving your revised submission.

Yours sincerely,

Kim Barrett
Senior Editor
The Journal of Physiology

REQUIRED ITEMS:

- Papers must comply with the Statistics Policy: https://jp.msubmit.net/cgi-bin/main.plex?form_type=display_requirements#statistics.

In summary:

- If $n \leq 30$, all data points must be plotted in the figure in a way that reveals their range and distribution. A bar graph with data points overlaid, a box and whisker plot or a violin plot (preferably with data points included) are acceptable formats.
- If $n > 30$, then the entire raw dataset must be made available either as supporting information, or hosted on a not-for-profit repository, e.g. FigShare, with access details provided in the manuscript.
- 'n' clearly defined (e.g. x cells from y slices in z animals) in the Methods. Authors should be mindful of pseudoreplication.
- All relevant 'n' values must be clearly stated in the main text, figures and tables.
- The most appropriate summary statistic (e.g. mean or median and standard deviation) must be used. Standard Error of the Mean (SEM) alone is not permitted.
- Exact p values must be stated. Authors must not use 'greater than' or 'less than'. Exact p values must be stated to three significant figures even when 'no statistical significance' is claimed.

Reviewing Editor's comments:

Thank you for your revision. Reviewer #1 has highlighted a few minor typographical errors and reference checking to be done. And in accordance with the ethics policy please could authors provide an ethics approval number.

Referee #2:

The authors have addressed my previous concerns and amended the text for better comprehension. They have also made necessary changes to the figures as previously suggested. My primary concern for this manuscript was the need to demonstrate the feasibility of the DECIDE technology for higher SO₂s, especially given that their previous papers have used DECIDE in a similar SO₂ range. However, after clarifying with the authors, I agree that showing DECIDEs sensitivity to detect hyperoxia in the same subject following maternal supplemental oxygen may be necessary if this therapy becomes clinical practice for FGR or PE. Minor comments below:

Please check for spelling and grammatical errors one more time. For example, in Lines 188-189, the publication year for Schrauben et al. is incorrect and is currently written as "20219"...please change it to 2021 or 2019. Line 431 spelling mistake/repeated word: "furter".

In Lines 411-412, it was my understanding that the human placenta is a concurrent exchanger...although this may still be an open question. The current consensus may be that it combines concurrent and countercurrent exchange streams. Please double-check the reference to make sure. There may be more updated modelling of human placental exchange that addresses this question.

Referee #3:

I thank the authors for attending to my concerns and addressing the limitations. Other than my belief that "advanced MRI techniques" is not an appropriate term, I acknowledge that, in the context of the current investigation, the term is acceptable.

I have no further comments.

END OF COMMENTS

We are grateful again to both reviewers for their comments and feedback. Detailed responses to Reviewer 1 are given below.

Referee #2:

The authors have addressed my previous concerns and amended the text for better comprehension. They have also made necessary changes to the figures as previously suggested. My primary concern for this manuscript was the need to demonstrate the feasibility of the DECIDE technology for higher SO₂s, especially given that their previous papers have used DECIDE in a similar SO₂ range. However, after clarifying with the authors, I agree that showing DECIDEs sensitivity to detect hyperoxia in the same subject following maternal supplemental oxygen may be necessary if this therapy becomes clinical practice for FGR or PE. Minor comments below:

1. Please check for spelling and grammatical errors one more time. For example, in Lines 188-189, the publication year for Schrauben et al. is incorrect and is currently written as "20219"...please change it to 2021 or 2019. Line 431 spelling mistake/repeated word: "furter"

R1.1. – We thank the reviewer for pointing this. We have corrected the publication year in lines 188-189. We have proofread the manuscript again.

Lines 185–187: *“The sensor was placed on the pregnant ewe’s teat and measurements were continuously recorded using LabChart 7 (AD Instruments, Bella Vista, NSW, Australia) (Darby et al. 2019; Duan et al. 2019; Schrauben et al. 2019).”*

Lines 423–424: *“In addition, the perfusion and oxygenation properties of different placentome types in the sheep is an area of future further study.”*

2. In Lines 411-412, it was my understanding that the human placenta is a concurrent exchanger...although this may still be an open question. The current consensus may be that it combines concurrent and countercurrent exchange streams. Please double-check the reference to make sure. There may be more updated modelling of human placental exchange that addresses this question.

We thank the reviewer for this comment. We have modified the sentence and added a more recent reference.

Lines 403–406: *“In addition, the sheep placenta is functionally different in that it is believed to be a concurrent hemodynamic system with more layers between maternal and fetal blood, which is less efficient than the counter current system evidenced in the human placenta (McNanley et al. 2008, Schroder et al. 1995).”*

Dear Dr Flouri,

Re: JP-TFP-2025-287272R2 "Feasibility of multi-modal magnetic resonance imaging to assess maternal hyperoxygenation in sheep pregnancy" by Dimitra Flouri, Jack RT Darby, Stacey L Holman, Georgia K Williams, Vasileios Vavourakis, Anna Louise David, Janna L Morrison, and Andrew Melbourne

We are pleased to tell you that your paper has been accepted for publication in The Journal of Physiology.

Authors should note that it is too late at this point to offer corrections prior to proofing. Major corrections at proof stage, such as changes to figures, will be referred to the Editors for approval before they can be incorporated. Only minor changes, such as to style and consistency, should be made at proof stage. Changes that need to be made after proof stage will usually require a formal correction notice.

All queries at proof stage should be sent to: TJP@wiley.com

If you would like to receive our 'Research Roundup', a monthly newsletter highlighting the cutting-edge research published in The Physiological Society's family of journals (The Journal of Physiology, Experimental Physiology and Physiological Reports), please click this link, fill in your name and email address and select 'Research Roundup':
<https://www.physoc.org/journals-and-media/membernews/>

Yours sincerely,

Kim Barrett
Senior Editor
The Journal of Physiology

P.S. - You can help your research get the attention it deserves! Check out Wiley's free Promotion Guide for best-practice recommendations for promoting your work at www.wileyauthors.com/eeo/guide. You can learn more about Wiley Editing Services which offers professional video, design, and writing services to create shareable video abstracts, infographics, conference posters, lay summaries, and research news stories for your research at www.wileyauthors.com/eeo/promotion.

IMPORTANT NOTICE ABOUT OPEN ACCESS: To assist authors whose funding agencies mandate public access to published research findings sooner than 12 months after publication The Journal of Physiology allows authors to pay an Open Access (OA) fee to have their papers made freely available immediately on publication.

You can check if your funder or institution has a Wiley Open Access Account here: <https://authorservices.wiley.com/author-resources/Journal-Authors/licensing-and-open-access/open-access/author-compliance-tool.html>

Reviewing Editor's comments to the authors:

Thank you for the amendments. There are no further questions to be addressed

END OF COMMENTS